# Reversible Splenial Lesion Syndrome as a Challenging Casuistry

**DOI:** 10.3390/ijerph19169842

**Published:** 2022-08-10

**Authors:** Dominika Tatar, Bogusława Bocian, Krzysztof Świerzy, Karina Badura Brzoza

**Affiliations:** Clinical Department of Psychiatry, Faculty of Medical Sciences in Zabrze, Medical University of Silesia in Katowice, Research Circle, 40-055 Katowice, Poland

**Keywords:** reversible splenial lesion syndrome, stroke, corpus callosum, schizophrenia

## Abstract

The corpus callosum plays a vital role in brain function. In particular, in the trunk of the corpus callosum, in the course of various diseases, there may be temporary, reversible changes (reversible splenial lesion syndrome (RESLES)), as well as partially reversible and irreversible changes. This article discusses the differentiation of RESLES and other conditions with changes in the corpus callosum lobe, as well as the accompanying clinical symptoms. Moreover, a case report of a patient in whom the above changes appeared in the nuclear magnetic resonance (NMR) image is presented. A 20-year-old patient with the diagnosis of Ehlers–Danlos syndrome type VI was admitted to the psychiatric ward in an emergency because of psychomotor agitation, refusal to take food and fluids, delusional statements with a message, grandeur, and auditory hallucinations. In the performed magnetic resonance imaging (MRI) of the brain, the corpus callosum non-characteristic in T2-weighted images revealed a hyperintensive area, which was significantly hyperintensive in diffusion magnetic resonance (DWI) sequences and in apparent diffusion coefficient (ADC) sequences with reduced signal intensity and no signs of bleeding. The hypothesis of subacute ischemic stroke of the corpus callosum was presented. In the control MRI of the brain, changes in the corpus callosum completely regressed, thus excluding an ischemic etiology and favoring the diagnosis of RESLES. During hospitalization, the patient experienced significant fluctuations in mental status, with the dominant symptoms typical of the paranoid syndrome in the form of disturbances in the course and structure of thinking and perception, and a clear and stable improvement was obtained after the administration of long-acting intramuscular olanzapine. Taking into account the clinical and radiological picture, the age of the episode, the rapidity of the disease development, the persistence of its clinical symptoms after the withdrawal of radiological changes in the brain NMR image, as well as the significant improvement in the clinical condition after the introduction of antipsychotic drugs, the final diagnosis was made of schizophrenia.

## 1. Introduction

The corpus callosum (Latin: corpus callosum) is the largest subcortical commissural fiber. It includes the trunk of the corpus callosum (Latin: truncus corporis callosi), the rostrum (Latin: genu corporis callosi), genu (Latin: rostrum corporis callosi), isthmus (Latin: lamina rostralis), and the most posterior and thickened part, the splenium (Latin: splenium corporis callosi) [1]. It has an important role in the functions of the brain. It creates a bridge between both hemispheres of the brain, containing intersecting axonal fibers, which, in the lobe of the corpus callosum, are projections from the occipital–parietal and temporal cortex [2,3]. Thus far, the function of the corpus callosum has not been fully understood, but studies have shown that its damage may result in the disconnection of the cerebral hemispheres, with disruption of higher cortical functions and disturbances in the stream of consciousness [3].

Changes in the splenium are described in many diseases—irreversible or partially reversible—and are related, among others, to isolated corpus callosum infarction (ICCI), head region injuries, congenital malformations (agenesis/dysgenesis of the corpus callosum, lipoma), metabolic disorders (Wilson’s disease, X-linked adrenoleukodystrophy, Krabbe’s disease), and neoplastic tumors (glioblastoma, lymphoma). Transient, reversible changes occur in viral, bacterial, and parasitic infections; during therapy with the use of metronidazole or 5-fluoro-uracil; in the case of toxicity or discontinuation of antiepileptic drug treatment, especially levetiracetam, carbamazepine, and valproates; as well as in states of hypoglycemia, hyper- or hyponatremia, and in high-mountain cerebral edema or in patients with Wernicki’s syndrome [2,4,5,6,7]. The clinical and radiological condition associated with these changes was defined as “reversible splenial lesion syndrome” (RESLES) [2,3]. The literature also includes the terms “mild encephalitis/encephalopathy with reversible splenium lesion” (MERS), “boomerang lesions”, and “corpus callosum diffusion limited cytotoxic lesions” (CLOCC). The term RESLES seems to be the most appropriate one because it emphasizes the reversibility of the changes and their localization [3]. Performing brain imaging with the use of magnetic resonance imaging is an important diagnostic tool visualizing the extent of changes within the corpus callosum. Imaging of the Central Nervous System (CNS), together with the data obtained from the patient’s medical history and physical examination, is important for the diagnostic and therapeutic process, and also allows the physician to establish the final diagnosis and select the appropriate treatment (Table 1).

The shape of the lesions, their location, and the intensity of the signal may help the physician to make a diagnosis. In the acute phase of RESLES, a local increase in signal intensity is seen in the DWI sequence, with a decreased signal on the ADC map consistent with cytotoxic edema. The abnormal region is hyperintense in T2 and fluid-attenuated inversion recovery (FLAIR) sequences and hypointense in T1, surrounded by normal axon crossing. The time of development and disappearance of changes in RESLES varies—in the case of infections, they are most often visible from one day and gradually disappear within 1–2 weeks, while, in the case of the toxic effect of an antiepileptic drug, they may persist for up to three weeks [3,7]. As mentioned, the differentiation should also take into account the edema of the corpus callosum (changes in the form of central rims, abdominal lesions in the splenium, hyperintensive DWI), Wilson’s disease (bilateral, symmetrical morphologically different in the splenium, and also involving the basal nuclei, thalamus, midbrain, bridge, and nucleus), teeth (which, in MRI, present as hyperintensive DWI, FLAIR, without hypointense ADC, white matter edema), X-linked adrenoleukodystrophy (polymorphic lesions including splenium, as well as white matter in the parieto-occipital area, manifesting as hyperintense T2, FLAIR, or FLAIR with variable enhancement of contrast), Krabbe’s disease (globoid cells in the splenium, cortico-spinal tract, and white matter of the parietal lobes, hyperintense axial T2, hypointense sagittal T1, FLAIR, DWI, ADC), glioblastoma multiforme (irregular large mass with infiltration and necrosis, including splenium, supratentorial white creature, hyperintense sagittal T1 and axial T2), B-cell lymphoma (irregular mass along the perivascular spaces, splenium that represents a hyperintensive center and hypointense periphery in the axial FLAIR, isointense T2, iso- or hypointense T1, DWI, ADC), lipoma (often accompanied by agenesis or dysgenesis of the corpus callosum, which, in the MRI image, is manifested by hyperintense T1 with a slight restriction of diffusion in DWI and ADC), post-traumatic changes (morphologically differentiated, asymmetrical with a midline shift, which, in addition to the history, may also be indicated by reduced diffusion of DWI and ADC), and intracranial hypotension (stump-like, downwardly displaced splenium, brain displacement, dilation of veins and sinuses, and its strengthening, isointense in FLAIR) [3,4,5,6,7,8,9,10].

The most common prodromal symptom of RESLES is fever, and the most frequently observed clinical picture is disturbance of consciousness with subsequent full-blown delirium and seizure, which disappear within a month [3,11,12]. Other symptoms include headache, disorientation, confusion, hallucinations and other psychotic symptoms, ataxia, dysarthria, disconnection syndrome, and coma [6,12,13,14,15]. Particular attention should be paid to the disconnection syndrome, also known as split-brain syndrome or dyscirculation syndrome, which is known, e.g., as a complication of calosotomy (a neurosurgical procedure involving cutting the corpus callosum, used mainly in severe forms of drug-resistant epilepsy) [16]; however, it may also be present in RESLES. In its course, the coexistence of several independent consciousnesses in one brain was observed, as well as visual integration disorders (each hemisphere independently perceives the contralateral field of view; however, proper functioning of the corpus callosum is necessary for integration; in the case of disturbance of its functions, the patient is not able to compare two objects when each of them is presented in a different field of vision—right and left), limited reception of stimuli from the non-dominant hemisphere, tactile and/or visual anomy (inability to identify previously known objects, concepts), alien limb syndrome, confusion, hallucinations, psychosis, motor aphasia, mutism, akinesia, and hemiplegia [3,6,12,13,14,15,16].

However, it should be borne in mind that these symptoms are non-specific and may appear in many other diseases, including those listed above. At the same time, not always in the case of their occurrence, MRI of the brain is performed, especially when the result of the computed tomography of the head is normal. Meanwhile, the available clinical trials have shown, among others, that in 5.71% of hypoglycemic patients and in 1.35% of patients with influenza experiencing intermittent episodes of disturbances of consciousness, there are changes characteristic of RESLES [3,13,14,15].

The aim of the study is to present the difficulties related to the diagnostic and therapeutic management of patients with impaired consciousness and to emphasize the role of brain imaging using magnetic resonance imaging.

## 2. Case Report

Man, 20 years old, BMI = 13.88, under the care of the Genetic, Cardiology and Orthopedic Clinic due to the suspicion of Marfan syndrome, and then Ehlers–Danlos syndrome type VI, caused by a homozygous mutation in the PLOD1 gene, from pregnancy I, delivery I with uncomplicated course. At birth, length 58 cm, body weight 2800 g, APGAR 6/9; physical examination revealed high articular laxity and clubfoot. Motor development was delayed; at the age of 10 months, MRI of the brain was performed, which revealed hypoxic–ischemic changes. Chest deformity was also observed, which worsened with age, while, in EMG examination, primary myopathy was excluded. In the following years, he was diagnosed with bilateral hyperopia with hyperopic astigmatism, and when he was 13 years old, the patient underwent amputation of the left lower limb at the level of the left thigh due to a rupture of the left popliteal artery aneurysm. The patient also had two spontaneous pneumothoraxes—for this reason, surgery was performed to remove the apex of the lung.

The patient was brought to the hospital emergency department (HED) by the medical rescue team, accompanied by his mother. The family reports showed that the patient had been behaving bizarrely for a week, with severe psychomotor agitation, psychotically motivated refusal to drink and drink, delusional statements with a message, grandeur, and auditory hallucinations. The symptoms were preceded by several weeks of anxiety and insomnia, which the family associated with his difficult personal situation—a close friend had attempted suicide, which had a strong negative impression on the patient. At the same time, the interview regarding the use of psychoactive substances and head injuries was negative. The day before the admission, the patient attacked a paramedic in the HED.

Due to the condition that the patient could not express his/her informed consent, he was admitted to the ward under Art. 22 sec. 2a UoOZP [17]. On admission: conscious, without logical contact, disoriented about the place and time, with psychomotor agitation, in an accelerated drive, unstable mood, with disturbed form and content of thinking in the form of disorganization, paralogy, delusional statements of a grandiose nature as well as religious and missionary ones without structured content. Additionally, echolalia and auditory and visual hallucinations were observed. A complete physical examination was impossible due to the patient’s lack of cooperation; the aberrations from the normal state revealed features of dehydration (drying up mucous membranes), deformation of the chest, condition after amputation of the left lower limb at the thigh level. A preliminary diagnosis of consciousness disorder of unknown etiology was made for differentiation from acute paranoid syndrome on the basis of schizophrenia.

The results of the obtained laboratory tests are presented in the Table 1 below.
ijerph-19-09842-t001_Table 1Table 1Laboratory tests performed on the patient.Hospitalization Day1237Reference ValuesWBC [×10^3^/µL]17.8117.2910.1012.424.00–10.00RBC [×10^6^/µL]5.835.044.674.874.50–5.90HGB [g/dL]18.616.415.315.813.5–17.5HCT [%]53.146.542.643.840.0–51.0PLT [×10^3^/µL]306257201254150–400Na+ [mmol/L]150.1151.1143.1142.9136–146K+ [mmol/L]4.313.663.503.683.5–5.1Glucose [mg/dL]9079--70–100Creatinine [mg/dL]0.70.730.340.540.67–1.17ALT [U/L]38.6--27.910.0–37.0AST [U/L]154.5--31.110.0–37.0CK-NAC [U/L]73918928612130525–175CRP [mg/L]17.40-9.829.900.00–5.00


On the first day of hospitalization, the use of the direct means of physical coercion was performed in the form of immobilization with magnetic belts and the patient was placed in an observation room, catheterized, and vital parameters were monitored. The treatment included haloperidol (5 mg/mL/d) and lorazepam (4 mg/d) in the form of i.m. injections and parenteral hydration. Control tests were performed, the results of which indicated a further increase in leukocytosis, hypernatremia, and an increase in creatine kinase levels. The concentration of troponin I was 21.30 [mg/L], CK-MB 64 [U/L], D-dimers 1801.37 [ng/mL]. The consulting internist excluded acute coronary syndrome and pulmonary embolism. No signs of bacterial or viral infection were observed. Computed tomography of the head showed no abnormalities; chest tomography showed large deformation of the chest with a significant reduction in the sagittal dimension and the wedge flattening of the L1 vertebrae with probable post-traumatic etiology. Enoxaparin 40 mg/0.4 mL s.c. was included in the treatment; parenteral hydration with 5% glucose solution was applied. From the third day of stay, these parameters were gradually stabilizing. Slow resolution of disturbances of consciousness and the presence of persecutory statements and delusional behavior, influence, and revealing thoughts, as well as magical thinking and associative association, resulted in the decision to modify the treatment. Olanzapine was used in the form of i.m. injection. The intensity and ferocity of the above-mentioned symptoms in the clinical picture and the appearance of disturbances of consciousness, despite the correct result of the computed tomography (CT) scan of the head, did not allow for the complete exclusion of the stroke process. MRI of the brain was performed with intravenous administration of a contrast agent, showing streaked, bilateral, T2-intense paraventricular areas with a predominance of changes on the left side, with a significant reduction in the volume of the left paraventricular white matter and distortion of the outline of the lateral ventricles—mainly the left—with possible hypoxic–ischemic etiology, and, in the lobe of the corpus callosum, a hyperintense area uncharacteristic in T2-weighted images, which was significantly hyperintense in DWI sequences and had reduced signal intensity in APC sequences.

No bleeding features were visualized in this area and a hypothesis of subacute ischemia within the 12 × 4 mm corpus callosum in the transverse plane was put forward. In view of the obtained results, the patient was consulted neurologically and an angio-CT examination of the intracranial arteries was performed, which revealed asymmetry of vertebral arteries with the dominant right artery, but no signs of vascular malformation were found. Due to the patient’s lack of cooperation, as well as unambiguous indications to perform the procedure, the lumbar puncture was abandoned.

In the course of further diagnostics, the patient was diagnosed with hypovitaminosis D3 (14.6 ng/mL), while antiphospholipid syndrome was excluded (level of p/cardiolipin antibodies in IgM and IgG classes < 2.00 U/mL, level of p/β-2-glycoprotein antibodies in IgM and IgG < 2.00 U/mL). The patient’s stay in the ward was complicated by an infection of the urinary tract with Escherichia coli ESβL on the 14th day of hospitalization. Pharmacotherapy was carried out in accordance with the antibiogram, improving the physical condition.

During the patient’s stay in the ward, due to the lack of cooperation, treatment was carried out with olanzapine in the form of intramuscular injections (up to 20 mg/day), and then, after the reduction of acute productive symptoms, in the oral form (up to 20 mg/day). After a few days, significant deterioration in mental state was observed (probably resulting from the patient not taking medication), in the form of intensified delusional content, mainly of a religious nature, hallucinations, and psychomotor agitation. The patient refused to take medication; auto- and allo-aggressive behaviors were present; therefore, temporary restraint in the form of immobilization with magnetic belts was used again. Pharmacotherapy was continued with the reintroduction of olanzapine (up to 20 mg/day) in the i.m. form, along with haloperidol (4 mg/day) in the form of an oral solution and additionally lorazepam (4 mg/day) in the event of significant agitation. Eventually, a marked improvement in mental state was achieved and it was decided to continue the treatment with intramuscular olanzapine with extended release (300 mg every 14 days). After 6 weeks of hospitalization, the patient underwent another MRI of the brain, which showed complete regression of changes in the corpus callosum, thus excluding ischemic etiology and favoring the diagnosis of RESLES. In addition, the MRI image of the brain was comparable to the previous examination. Despite the complete withdrawal of changes in the NMR image, clinically, the patient showed a slight paralogy in thinking, a tendency toward magical thinking, and disturbing statements. In his behavior, however, he was calm and cooperative, with superficial criticism of the experienced psychotic symptoms and a lack of a sense of illness. The continued antipsychotic treatment allowed the patient to be discharged home in a state of significant improvement.

On the day of discharge, the patient was clearly aware, comprehensively oriented, calm and adjusted in behavior, did not seem to be hallucinating, and did not spontaneously express delusional content. Taking into account the entire clinical picture (with the dominant productive symptoms characteristic of the endogenous process) and the results of the research, the diagnosis of paranoid schizophrenia was made and it was recommended to continue pharmacotherapy on an outpatient basis under ambulatory care.

## 3. Discussing

RESLES is a rare condition with a wide clinical and radiological spectrum occurring in the course of many diseases, metabolic disorders, and other disease states [3,4,5,6,7,12,18]. In most cases, clinical symptoms tend to improve or even completely resolve; however, in patients with severely disturbed consciousness, the prognosis is unfavorable [13,14,15,18]. In the case presented above, in the performed CT scan of the head, no deviations were found. Due to the data obtained from the history of the sudden onset of symptoms and their duration, nonspecific laboratory test results, and a severe clinical course with the presence of disturbed consciousness, it was decided to perform a head MRI, which showed changes in the corpus callosum characteristic of a previous ischemic stroke [2,4,8]. Previous studies were not available for comparison; therefore, based on the morphology, distribution, and coexistence of changes in the paraventricular white matter and the clinical picture, a hypothesis about a history of ischemic stroke was made [4,18]. On the other hand, the control MRI showed complete regression of changes in the corpus callosum, which gave grounds for the diagnosis of RESLES [3,4,5,7,8,9,11,18]. In the clinical picture, at the same time, there was no complete withdrawal of positive symptoms—typical for schizophrenia, there were distracted logical thinking disorders, the occurrence of hallucinations and delusions with the content characteristic of the above-mentioned disease entity, disorganization and bizarre behavior, psychomotor agitation, and no criticism of one’s own judgments. The above-mentioned symptoms started to improve only after intensive antipsychotic treatment.

Taking into account all the available information, it seems that, in the described case, symptoms typical of the schizophrenia picture and symptoms related to reversible changes in the corpus callosum (RESLES) could have coexisted. It is difficult to determine what could have been the cause of the above-mentioned changes and whether they could have influenced the clinical condition of the patient, which, in the initial period of hospitalization, seemed to indicate an acute cerebral cause. The high levels of sodium recorded in laboratory tests could be considered a probable cause of changes in the corpus callosum, but this is only a hypothesis. The cause of electrolyte disturbances was not found either. A family interview indicated the probability of the presence of prodromal symptoms of schizophrenia. A few weeks before hospitalization, the patient began to show changes in behavior, with some unusual behavior and suspicion in his speech. During this period, there were no symptoms of a disturbed consciousness [15]. The complete withdrawal of changes in the NMR image, the presence of symptoms typical of the picture of schizophrenia (meeting the ICD-10 criteria for this disease entity), the age at the episode’s onset, the rapidity of the disease development, and the improvement in the clinical condition after the initiation of antipsychotic drugs resulted in the final decision to make the diagnosis of schizophrenia excluding the diagnosis of organic psychotic disorders [15,19]. Taking into account the final diagnosis in differentiation, one should consider the following.

1. Acute psychotic disorders caused by the use of psychoactive substances

In the course of the use of psychoactive substances, the patient may develop perception disorders in the form of hallucinations, delusions, psychomotor agitation, aggressive behavior, disturbances in the rhythm of sleep and wakefulness, as well as anxiety, restlessness, or affective disorders. In the described case, this diagnosis could be ruled out due to a negative history of addiction [20,21].

2. Organic psychotic disorders

As mentioned above, psychotic disorders may appear on the basis of various organic changes in the CNS (resulting from, among others, brain tumors, bacterial and viral diseases, epilepsy, porphyria, or intoxication with drugs and heavy metals). However, it should be remembered that, in the discussed case, there was a complete withdrawal of changes in imaging tests, with the continued presence of clinical symptoms, which allowed for the rejection of the above diagnosis. Moreover, the patient had no signs of infection [22].

3. Psychotic disorders in the course of affective disorders—severe depressive episode with psychotic symptoms, psychotic mania

In the course of affective disorders, the contents of delusions and hallucinations are adjusted to the mood, which was not observed in the patient [23].

4. Delusional disorders

In their course, delusions are systematized, without the coexistence of hallucinations, affect disorders, forms of thinking, and social maladjustment. This diagnosis can also be excluded in a patient due to the presence of hallucinations and disturbances in the structure of thinking [23].

5. Post-traumatic stress disorder

It is difficult to argue that the stress factor did not significantly affect the mental state of the patient; however, the clinical symptoms did not meet the criteria for the diagnosis of an F43 disorder [24], which include ruminescences, i.e., episodes of repeated intrusive thoughts and ideas of a threatening or catastrophic nature and concomitant anxiety. Additionally, it may be accompanied by indifference, apathy, or anhedonia.

## 4. Conclusions

Carrying out a detailed and reliable diagnostic process, including radiological examinations, allows for the correct final diagnosis. The limitations of computed tomography and magnetic resonance imaging should be borne in mind when considering the use of both techniques for diagnosis. It should also be taken into consideration that psychotic disorders can occur in many disease entities. Moreover, the cooperation of clinicians seems to be extremely important, allowing the differentiation to be carried out with the exclusion of suspected disorders.

## Data Availability

Not applicable.

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
