# Peer review of "Reversible Splenial Lesion Syndrome as a Challenging Casuistry"

_ijerph, 2022, doi:10.3390/ijerph19169842_

Round 1

Reviewer 1 Report

This article discusses the differentiation of Reversible splenial lesion syndrome and other conditions that proceed with changes in the corpus callosum accompanied by the clinical symptoms. The authors described a patient case report.

I have no doubts about the importance of the topic, but the article must be written following the international guidelines that provide the tool to inform and simplify the process of writing accurate and transparent case reports. https://www.care-statement.org/

Author Response

Dears Sirs

Thank you for revising article. We have reorganized the paper following Reviewers’ comments. All changes in the text of article have been written in red pencil. There are our responds to the Reviewers.

  1. We provided sufficient background to the introduction provide and include all relevant references

  2. We reorganized the cited references.

  3. We reorganized the resarch design according to international guidelines that provide the tool to inform and simplify the process of writing accurate and transparent case reports. https://www.care-statement.org/

  4. We tried to describe methods adequately.

  5. We improved the presentation of the results to be more clear for readers.

  6. We reconstructed conclusion

Reviewer 2 Report

Thank you for sending the manuscript for review. I believe that before final decision, the manuscript needs substantial revisions. My comments:

1. The title should be rewritten in a more scientific manner. It`s like title of a article in a magazine. In addition, any abbreviation should be avoided in the title.

2. Any abbreviation should be used in full-term for first time.

3. Course and treatment of the case should be described more in the abstract.

4. 'RESLES' should be replaced with an appropriate key-word from MeSH.

5. Anatomical descriptions should be summarized in the introduction. In addition, I don`t think table-1 is appropriate for a case report and I recommend removing it.

6. What does it mean: 'religiously motivated refusal to drink and eat'? or 'which the family associated with his difficult personal situation (a friend's suicide attempt)'.

7. Combination of Lorazepam and Olanzapine is not allowed and some references state it as a contraindication. It has been prescribed for the patient. Why?

8. Differential diagnoses should be more discussed.

9. Conclusion and practical implications should be added.

Author Response

Dears Sirs

Thank you for revising article. We have reorganized the paper following Reviewers’ comments. All changes in the text of article have been written in red pencil. There are our responds to the Reviewers.

1. The title should be rewritten in a more scientific manner. It`s like title of a article in a magazine. In addition, any abbreviation should be avoided in the title.

ad. 1

Corrections mentioned above were made in the text of the article-

2. Any abbreviation should be used in full-term for first time.

ad. 2

Corrections were made to the text of the article

3. Course and treatment of the case should be described more in the abstract.

ad. 3

Course if the treatment have been described more in the abstract

4. 'RESLES' should be replaced with an appropriate key-word from MeSH.

ad. 4

Corrections were made to the text of the article

5. Anatomical descriptions should be summarized in the introduction. In addition, I don`t think table-1 is appropriate for a case report and I recommend removing it.

ad. 5

Corrections were made to the text of the article

  1. What does it mean: 'religiously motivated refusal to drink and eat'? or 'which the family associated with his difficult personal situation (a friend's suicide attempt)'.

    We corrected these sentences

  1. Combination of Lorazepam and Olanzapine is not allowed and some references state it as a contraindication. It has been prescribed for the patient. Why?

    Lorazepam were administrated to the patients only ad hoc in the state of extremally excitation.

  1. Differential diagnoses should be more discussed.

    We have disscused it

  1. Conclusion and practical implications should be added.

We added it

Round 2

Reviewer 1 Report

Thank you for your replies!

Author Response

Dears Sirs

Thank you for revising article. 

Kind regards

Reviewer 2 Report

Thank you for submission of the revised manuscript. Only thing is lack of appropriate proof-reading. There are several typos throughout the text such as 'MRI (MRI)',  'auditory hallucinations, and auditory hallucinations.'. Content of the manuscript is appropriate by and large and my comments have been addressed in a good way.

Author Response

Dears Sirs

Thank you for revising article. We have reorganized the paper following Reviewers’ comments. All changes in the text of article have been written in red pencil.